# Timing Is Everything: The Metabolic Partitioning of Suberin-Destined Carbon

**DOI:** 10.3390/plants14101433

**Published:** 2025-05-10

**Authors:** Jessica L. Sinka, Mark A. Bernards

**Affiliations:** Department of Biology, Western University, London, ON N6A 5B7, Canada; jsinka2@uwo.ca

**Keywords:** suberin, wound healing, carbon partitioning

## Abstract

Suberin is a cell wall-associated biopolymer that possesses both poly(phenolic) and poly(aliphatic) elements assembled into chemically and spatially distinct domains. Domain-specific monomers are formed via a branched pathway between phenolic and aliphatic metabolisms. Previous transcript accumulation data (RNAseq) from early stages of wound-induced suberization revealed highly coordinated, temporal changes in the regulation of these two branches. Notably, phenolic metabolism-associated transcripts accumulated first, indicating a preference toward phenolic production early on post-wounding. To better understand the dynamics of suberin monomer biosynthesis and assembly, we assessed carbon allocation between phenolic and aliphatic metabolisms during wound-induced suberization. To do so, [^13^C_6_]-glucose was administered to wound-healing potato tuber discs at different times post-wounding, and patterns of heavy carbon incorporation into (1) primary metabolites and (2) the suberin polymer were assessed. During early stages of wound-healing, carbon from glucose was rapidly incorporated into phenolic-destined metabolites, while at later stages it was shared between phenolic- and aliphatic-destined metabolites. Similarly, the pattern of labelled carbon incorporation into the poly(aliphatic) domain reflected a greater dedication of carbon towards 18:1 w-hydroxy fatty acid and 18:1 dioic acid (the two most abundant aliphatic monomers in potato suberin) later in the wound healing time course.

## 1. Introduction

Wound healing in plants is an integral part of their overall defence system. The inducible nature of wound healing allows for a convenient way to probe metabolism, which is also associated with normal growth and development. For instance, the process of suberization, in which the biopolymer suberin is synthesized and deposited in cells, is initiated post-wounding in cells adjacent to the wound site, but is also a developmentally regulated process in specialized cells in below-ground tissue (e.g., root epidermis and endodermis and tuber phellem) and in above-ground tissue (e.g., bark and seed coats) [1,2]. The induction of suberization by wounding provides a clear starting point for the process, and wounded tissue is amenable to the application of a variety of exogenous metabolic precursors, including stable- and radioisotopes of important intermediates [3].

Structurally, *suberized* tissues contain both a polymeric phenolic component (i.e., poly(phenolic) domain; PPD) and the largely aliphatic polyester with associated waxes (i.e., poly(aliphatic) domain; PAD). *Suberin* has been variously described as a heteropolymer comprising two distinct, but covalently linked, domains (i.e., PPD and PAD) [1,2] and a polymer with alternating bands of aliphatics and phenolics [4,5]. Despite ambiguity on the exact macromolecular structure and composition of suberin, the chemical composition of suberized cell walls is well defined. Herein, we assume that the PPD of suberin is a covalent network of hydroxycinnamic acids and their derivatives, (including tyramine-derived hydroxycinnamic acid amides, and a lesser proportion of monolignols), embedded in the primary cell wall [6,7,8] and covalently linked to the PAD (which comprise the suberin lamellae made of fatty acids, w-hydroxy fatty acids, dioic acids, 1-alkanols, esterified hydroxycinnamic acids, and glycerol) as one complete polymer. Also found in the lamellae are alkyl ferulates, and associated waxes [9,10,11,12,13,14].

In the potato tuber, carbon derived from starch degradation supplies downstream metabolism, including the generation of PPD and PAD monomers via divergent pathways. The necessary precursory steps for the generation of PPD and PAD during suberization are based on conceptual knowledge and the logical biochemical sequence of metabolic steps [6,15,16]. Branching into disparate metabolic fates is initiated following the conversion of phosphoenolpyruvate (PEP), the penultimate molecule in glycolysis. One fate of carbon from PEP is the shikimic acid pathway, resulting in synthesis of L-phenylalanine (L-phe). L-phe is then converted into cinnamate by L-phe ammonia lyase (PAL), which is the first committed step of phenylpropanoid metabolism [17]. Phenylpropanoid-derived monomers (e.g., ferulic acid, feruloyltyramine, monolognols) are directly incorporated as the phenolic components of the suberized cell wall [18,19]. Alternatively, carbon from PEP can be converted into pyruvate and imported into mitochondria for oxidation via the tricarboxylic acid (TCA) cycle. Some pyruvate can support the production of suberin aliphatics via acetate exported from mitochondria and used to generate palmitic acid (C16:0) and stearic acid (C18:0) in plastids. These fatty acids are then modified via desaturation and oxidation to yield w-hydroxy fatty acids and dioic acids and/or elongation (via acetyl-CoA derived from citric acid in the cytoplasm) to yield very long chain fatty acids; the latter can be reduced to 1-alkanols or oxidized to yield longer chain (saturated) w-hydroxy fatty acids and dioic acids [20,21]. Modified fatty acids are then polymerized to other aliphatic monomers or to the phenolics in the primary cell wall through ester linkages with ferulic acid [12] or glycerol [10,14]. The PPD and PAD differentially protect exposed tissue at the site of wounding from infection by pathogens. For example, the PPD provides resistance to bacterial pathogens such as *Erwinia carotovora* (now *Pectobacterium carotovora*). Conversely the PAD confers the final resistance to the fungus *Fusarium sambucinum* [22]. Moreover, protection against water loss is conferred by proper formation of both PPD [23] and PAD [24,25,26], as recently shown for Arabidopsis endodermal tissue, in which PPD deposition is a prerequisite to aliphatic lamellae formation [27].

The two-domain model for potato suberin posits that the phenolic domain is synthesized and deposited in advance of aliphatic monomers, in a temporal sequence supported by physico-chemical [28,29], histochemical [30], metabolic [31], and transcriptomic [16] data. For example, some of the earliest histochemical analyses of suberized tissues revealed a lag in the detection of suberin aliphatics behind that of phenolics [30,32,33,34,35,36]. These initial studies gave rise to the idea of differential temporal regulation between biosynthesis of phenolic and aliphatic precursors and the resultant timing of their deposition [6]. Similarly, untargeted and targeted metabolic analyses, conducted on a potato tuber, also revealed temporally distinct metabolic profiles during wound metabolism arising from the differential induction of polar and non-polar metabolism, e.g., [26,31,37] with non-polar metabolites lagging behind polar metabolites. Efforts have also been made into identifying regulatory cues of the dynamics of phenolic and aliphatic metabolism. For example, de novo accumulation of ABA was induced by wounding and required for normal suberization. Specifically, when ABA biosynthesis was inhibited, there was a delay in expression of genes associated with aliphatic metabolism, but with no impact on genes for phenolic metabolism. From establishing ABA as an important regulator of aliphatic metabolism it can be surmised that its normal accumulation pattern, reaching a maximum 3 days post-wounding, is responsible (at least in part) for differential induction of phenolic and aliphatic metabolism. Many studies have shown that the expression of suberin aliphatic genes occurs only after expression of genes associated with suberin phenolics. Gene expression analysis has been conducted via RT-q-PCR using target specific primers [22,38,39], and more comprehensively by RNAseq [16]. Consistently, genes associated with phenylpropanoid metabolism, such as PAL and key hydroxylases and *O*-methyltransferases (OMT) integral in the conversion of *p*-coumaric acid into other hydroxycinnamic acids found in suberin, are expressed prior to aliphatic genes. Moreover, genes involved in phenolic assembly, such as anionic peroxidase, have also been shown to have earlier expression than those involved in aliphatic assembly, such as glycerol-3-phosphate acyltransferase (GPAT). The esterification of phenolics to aliphatics also exhibits a lag in expression, such as HXXXD family acyltransferases [16], presumably driven by the lag in production of the necessary aliphatic monomers. Overall, few studies have assessed the phenolic domain of suberin, be it precursory compounds or poly(phenolics), in wound-healing potato tubers. Even fewer have also probed polar- and non-polar metabolism overtime throughout closing layer formation. To address this knowledge gap, there has been increased interest into the biosynthesis and assembly of the phenolic domain. For example, to assess the effect of down-regulation of FHT on suberin formation, Jin et al., 2018 [26] measured polar and non-polar metabolites from potato native and wound-periderms. Distinct polar profiles for FHT-RNAi knockdown lines, implicating FHT (and thus active phenylpropanoid biosynthesis) early on in the wound-healing process was demonstrated [26].

However, previous efforts were still reliant on assumptions that an increase in gene expression will have direct metabolic consequences. Rather, the level of gene expression should be interpreted as having the ‘potential’ to effect metabolism. To this end, metabolic data are based on assessment of the abundance of analyte in the tissue at any given time and do not account for metabolic flux (i.e., rate of turnover). Due to the nature of metabolic data based on total abundance of metabolites, whether pools of free metabolites are incorporated into the final suberin polymer has not been confirmed. Isotopic labelling of tissues, via specific precursor compounds, is advantageous to non-isotope-based metabolic analyses for quantifying the balance between formation and utilization of metabolites, branching between pathways, identifying activated and inhibited reactions [40], and assessing end products of pathways. Flux can be derived from the rate in which isotopic label initially appears in downstream metabolites before the isotopic signature subsequently ‘disappears’ as the labelled metabolites are further metabolized. Here, we present a more generalized labelling approach where instead of calculating ‘flux’, we assess the dynamics of the total abundance metabolite pool over time in tandem with proportion of isotope enrichment. The predicted activity of metabolic steps can thus be concluded.

During wound-induced suberization in potato tubers, starch is likely the sole donor of carbon to downstream metabolites of interest; thus, we tracked the incorporation of [^13^C_6_]-glucose into downstream metabolism. To assess the incorporation of ^13^C into phenolic and aliphatic metabolisms, a variety of metabolic proxy compounds representative of each pathway were chosen to monitor various stages of suberin monomer biosynthesis. For example, shikimic acid and L-phe represent carbon upstream of phenolic metabolism, while citric acid, palmitic acid, and stearic acid represent carbon upstream of aliphatic metabolism. Using gas chromatography–mass spectrometry, we tracked the temporal partitioning of carbon between phenolic-destined monomers and aliphatic-destined monomers during wound-induced suberization. The incorporation of ^13^C label into the suberin polymer was also assessed. We report that during the initial 72 h post wounding (hpw), there was a clear temporal difference in the incorporation of administered label into phenolic and aliphatic metabolic pathways. Specifically, during early stages of wound-healing, carbon from glucose was rapidly incorporated into phenolic-destined metabolites, while at later stages, it was shared between phenolic- and aliphatic-destined metabolites.

## 2. Results

### 2.1. The Model System

The emergence of chromatography in tandem with mass spectrometry as a routine technology has allowed for estimation of stable isotope label incorporation into metabolites in various biochemical pathways. To assess the branched metabolic pathway that leads to production of suberin monomers, we used the well characterized wound-healing tuber model system, e.g., [7,16,24,36], since it (1) can be easily infiltrated with isotopic label at early wound healing timepoints, (2) is inducible allowing us to probe the “start” of these metabolic processes, and (3) produces enough suberized tissue for metabolite analysis. The stable isotope label was applied at 0, 24, 48 and 72 h post wounding to capture the hypothesized shift from phenolic to aliphatic metabolism, as reflected in metabolic [31] and transcriptional [16] changes induced by wounding, and second, they precede closing layer formation and therefore are conducive to [^13^C_6_]-glu application.

### 2.2. Wound Healing Is a Highly Dynamic and Temporal Process

It takes about 7 days for a suberized closing layer to form [41] under experimental conditions. To assess the process of suberin monomer production during this process metabolites were extracted in 1:3:1 MeOH-*tert*-butyl ether-H_2_O, and the polar-phase metabolites derivatized and run in the GC-MS. Several relevant metabolites were identified from the resulting chromatograms by retention time, mass spectra, and true standards (Figure 1).

Of note, the most abundant metabolites in the chromatograms were glucose and sucrose, which represent substantial pools of carbon available for downstream metabolism. The identified metabolites serve as proxies of the activity of carbohydrate metabolism (glucose and sucrose), the shikimic acid pathway (shikimic acid), phenylpropanoid biosynthesis (L-phe), the TCA cycle (citric acid), and fatty acid biosynthesis (stearic and palmitic acids). Metabolite abundance (Figure 2) represents a static measure of metabolite quantity at that moment in time, and the rate of turnover (i.e., balance between production and consumption) cannot be deduced.

In the case of citric acid, for example, its depletion overtime may be due to any dynamic where the source < sink. For example, citric acid may be being rapidly produced but more rapidly consumed or there is no new citric acid entering the pool, and thus it is being depleted as it is metabolized. In general, the data in Figure 2 indicate that, immediately post wounding, the tuber enters a state of metabolic rearrangement, where various compounds associated with wound–suberin production are disrupted from their steady state, before reaching a new steady state following wounding. Notably, compounds such as sucrose and shikimic acid varied in abundance over 7 days. Proxies for downstream suberin metabolism, such as L-phe, palmitic acid, and stearic acid, however, showed more consistency in their levels of abundance over time, though palmitic acid and stearic acid appeared to increase in abundance towards 168 hpw. The TCA intermediate citric acid, which also is the upstream precursor to the fatty acid elongation associated with aliphatic suberin formation, was depleted overtime.

### 2.3. Administered [^13^C_6_]-Glucose Is Taken Up and Rapidly Metabolized

The breakdown of glucose through carbohydrate metabolism feeds carbon towards key metabolites that have been implicated in suberin biosynthesis. [^13^C_6_]-glucose was selected as the label and applied immediately post-wounding to suberized tissue. Labelled tissue and control (non-labelled tissue) were sampled at 24 h intervals up to 72 h (Figure 3).

Infiltration of [^13^C_6_]-glucose resulted in approximately 41% ^13^C:^12^C glucose in the tissue, which was rapidly metabolized by 24 hpw (Figure 3a; Appendix A: ANOVA; F_3,16_ = 430, *p* < 0.0001). Finally, ^13^C was rapidly incorporated into downstream metabolites, such as sucrose. Following initial depletion of ^13^C from the glucose pool, by 72 hpw, there was a significant increase is the proportion isotopic label in glucose. Some carbon from the initial [^13^C_6_]-glucose was rapidly shunted directly into sucrose (Figure 3b), where it likely acted as a storage metabolite. The proportion of label incorporated into sucrose varied across early wound healing (Appendix A: one-way ANOVA; F_3,16_ = 8.7, *p* = 0.0012). Specifically, the most isotopic label was incorporated into sucrose at 24 hpw compared to immediately post wounding, 48, and 72 hpw (Figure 3b). Being that sucrose is either the most or second most abundant metabolite found in the polar metabolite extracts, it is likely to be a significant pool of carbon that can be drawn from at later timepoints during wound healing.

### 2.4. Phenolic Proxies Are Enriched with ^13^C Label in Early Wound Metabolism While Aliphatic Proxies Are Not

To assess turnover of metabolites over time, the dynamic of their accumulation, depletion, or steady state of metabolites (Figure 4) was interpreted with the proportion of the metabolite pool with ^13^C label (derived from a single bolus of [^13^C_6_]-glucose application). For tissue collected at the time of wounding (i.e., less than 0.5 h elapsed post wounding), the label was already incorporated into shikimic acid, a proxy for phenolic metabolism (one-way ANOVA; F_4,20_ = 2.1, *p* = 0.1136). The proportion of label incorporation into shikimic acid and other ‘proxies’ was greatest at 24 hpw (Figure 5).

At 24, 48, and 72 hpw, a label was found in citric acid, shikimic acid, and L-phe (one-way ANOVA; 24 hpw: F_4,20_ = 41, *p* < 0.0001, 48 hpw: F_4,20_ = 90, *p* < 0.0001, 72 hpw: F_4,20_ = 40, *p* < 0.0001), indicating that these molecules were being actively synthesized (Figure 5). While both palmitic and stearic acids exhibited similar steady-state levels in their total abundance in tissue (Figure 4), they were not isotopically enriched (Figure 5); this suggested that they were neither actively synthesized nor metabolized during this time frame. In the case of shikimic acid, which accumulated in the tissue over time (Figure 4), the proportional and relative amount of ^13^C label incorporation was negligible at 0 hpw (one-way ANOVA; 24 hpw: F_4,20_ = 2, *p* < 0.01372). After 24 hpw, ^13^C incorporation into the metabolite pool decreased but shikimic acid and L-phe remained the most ^13^C enriched metabolites (one-way ANOVA; 24 hpw: F_4,20_ = 29, *p* < 0.0001, 48 hpw: F_4,20_ = 61, *p* < 0.0001, and 72 hpw: F_4,20_ = 49, *p* < 0.0001) indicating ongoing synthesis. L-phe abundance in tissue decreased over time (Figure 4) while label continued to be incorporated into the L-phe pool (Figure 5). This likely means that L-phe was being consumed at a greater rate than it was being produced. Pairwise comparisons among proxies are presented for each sampling timepoint for proportion of pool with ^13^C label Appendix A and amount of ^13^C-enriched compound Appendix A.

Overall, quantifiable, and statistically relevant amounts of ^13^C label were identified in glucose and sucrose at all sampling timepoints based on one-sample *t*-tests in which the proportion of ^13^C label incorporation was tested against a theoretical mean of 0 Appendix A. One sample *T*-tests performed for the proxy metabolites indicated that in tissue collected immediately following wounding (<0.5 h) ^13^C label incorporation was statistically insignificant for all downstream proxies. Label was later confirmed in citric acid, shikimic acid, and L-phe at 24, 48, and 72 hpw. Palmitic acid only accumulated statistically relevant amounts of label 24 hpw, while stearic acid did not accumulate any ^13^C label at all timepoints Appendix A. Despite this confirmation, the proportion identified was less than initially anticipated given the amount available in glucose. This is likely because suberin monomers were not the sole metabolic sink for carbon derived from carbohydrate metabolism following wounding. Other metabolic carbon sinks upregulated by wounding include amino acid biosynthesis, sterol biosynthesis, carbohydrate metabolism, acetyl-CoA metabolism, lipid metabolism, the oxidation reduction process, brassinosteroid biosynthesis, indoleacetic acid biosynthesis, and isoprenoid biosynthesis [16].

### 2.5. Multiple Labelling Events with [^13^C_6_]-Glucose Were Rapidly and Similarly Metabolized

A single bolus application of [^13^C_6_]-glucose was limited by the rapid depletion of the [^13^C_6_]-glucose pool, leaving less available label for continuous integration over the 72 h time course. To correct for this, a ‘multi-bolus’ approach was adapted in which new isotopic label was applied to a subset of un-labelled wound healing tissue at the time of wounding as well as at 24, 48, and 72 h post-wounding. This ensured that there was substantial new label available to be incorporated into downstream metabolites and provided a unique opportunity to track the destiny of carbon derived from glucose into the suberin polymer from various timepoints during early wound healing. First, the rate of glucose turnover was assessed by sampling tissue at the time of labelling and 1, 3, and 6 h post labelling. Overall, more [^13^C_6_]-glucose was incorporated into the tissue at later timepoints than immediately post wounding. Generally, about 50% of the applied label was depleted within 3 h of application, regardless of how much was initially taken up by the tissue. The rate of proportional label depletion from glucose was similar among 24, 48, and 72 h labelling events. By contrast, the depletion of the label from the glucose pool labelled at 0 hpw was significantly slower (Appendix A, Figure 6, Simple linear regression: F = 5.425, DFd = 216, *p* = 0.0013).

The difference in the rate of depletion of the glucose pool likely reflects a brief lag in the level of metabolism immediately post wounding as genes governing wound-related processes are upregulated. ^13^C Label incorporation was calculated for proxy metabolites extracted 3 h after [^13^C_6_]-glucose administration since this was sufficient time for the administered [^13^C_6_]-glucose to be metabolized, with the proxy labelling still reflecting the metabolism at approx. 0, 24, 48, and 72 hpw. ^13^C label was still present in statistically relevant amounts up to and including 6 h post label application based on one-sample *t*-tests in which the proportion of ^13^C label incorporation was tested against a theoretical mean of 0 Appendix A.

### 2.6. Multiple Labelling Events Allow for Enhanced ^13^C Signal Intensity at Later Timepoints

More label was incorporated into the suberin proxy metabolites after each [^13^C_6_]-glucose application, especially at 24 hpw and onwards, than with a single bolus application at 0 hpw. Sucrose contained the highest proportional amount of ^13^C immediately following wounding (Appendix A, one-way ANOVA; F_3,50_ = 11, *p* < 0.0001), though there was more ^13^C-enriched compound relative to the size of the sucrose pool at later timepoints (Figure 7).

Thus, sucrose was likely being turned over into downstream metabolites at early timepoints at a rate at least equal to it being produced, while at later timepoints, it was being utilized at a rate less than it was being produced. Significant label was rapidly incorporated into citric acid, shikimic acid and to a lesser extent L-phe (i.e., within 3 hpw) following label application at 0 hpw. For both palmitic and stearic acids, total abundance remained low, with a relatively small proportion being isotopically enriched. This pattern of enrichment reflects an earlier onset of metabolism leading to polar metabolites relative to non-polar ones. Similarly, total shikimic acid increased with time, while ^13^C label incorporation peaked at 24 hpw (Figure 8). This pattern of label enrichment implied that early on, shikimic acid was rapidly turned over, but later, the flux through this metabolite slowed. This conclusion is supported by the pattern of label incorporation into L-phe, which peaked 48 hpw (Figure 8). Total L-phe in the tissue, however, decreased overtime (Figure 9), suggesting it was actively being synthesized but more rapidly utilized.
Figure 7^13^C-Label incorporation into sucrose following independent applications of [^13^C_6_]-glucose at different times post wounding. [^13^C_6_]-glucose was applied to tissue at the time of wounding and unlabelled wound-healing tissue 24, 48, and 48 hpw and the labelled glucose allowed to metabolize for 3 h post-application. (**a**) The amount of labelled compound relative to the total amount of metabolite present in the tissue (ug/mg tissue) (see Figure 9), and (**b**) the proportion of sucrose pool enriched with ^13^C label are presented. The amount of labelled compound was calculated by multiplying proportion of metabolite pool with ^13^C label by the total amount of sucrose in the sample at the time of sampling (See Figure 9). Replicates were normalized to the average proportion of [^13^C_6_]-glucose across all labelling timepoints. Data were analyzed by one-way ANOVA, followed by Tukey’s multiple comparisons test. Different letters above bars indicate significant differences (*p* < 0.05). Bars represent mean ± SEM (n = 10–15).
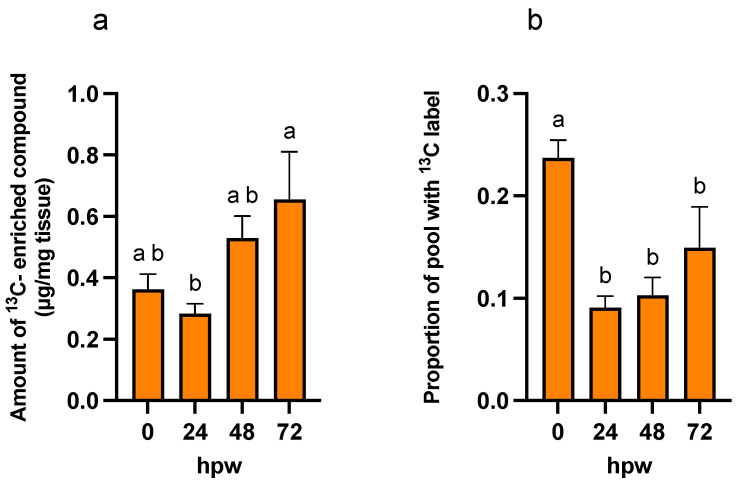



Overall, phenolic and aliphatic proxy metabolites differentially accumulated ^13^C as early as 0 hpw (one-way ANOVA; F_4,57_ = 6.3, *p* = 0.0003). This was most notable with 24, 48, and 72 hpw administered [^13^C_6_]-glucose, where L-phe had a greater proportion of ^13^C label incorporated than either palmitic or stearic acid (one-way ANOVA; 24 hpw: F_4,61_ = 46, *p* < 0.0001, 48 hpw: one-way ANOVA; F_4,61_ = 12, *p* < 0.0001, 72 hpw: one-way ANOVA; F_4,56_ = 22, *p* < 0.0001). Thus, carbon from glucose was preferentially directed to phenolic precursor metabolism within the first 72 h post wounding and not towards aliphatic precursor production. Importantly, this metabolism was rapidly fueled with upstream carbon (Figure 8 and Figure 9). Pairwise comparisons among proxies are presented for each sampling timepoint for proportion of pool with ^13^C label Appendix A and amount of ^13^C-enriched compound Appendix A. In general, ^13^C enrichment relative to total amount of proxy exhibited similar trends with citric acid and shikimic acid, but because there is little L-phe in the tissue across all sampling timepoints, it did not differ in total enrichment from palmitic or stearic acids (one-way ANOVA; 0 hpw F_4,57_ = 10, *p* < 0.0001, 24 hpw: F_4,61_ = 50, *p* < 0.0001, 48 hpw: F_4,61_ = 15, *p* < 0.0001, 72 hpw: F_4,56_ = 26, *p* < 0.0001). Overall, quantifiable and statistically relevant amounts of ^13^C label were identified in all proxies of suberin biosynthesis, except for stearic acid, following [^13^C_6_]-glucose application at 72 hpw, based on one-sample *t*-tests in which the proportion of ^13^C label incorporation was tested against a theoretical mean of 0 Appendix A.

### 2.7. Carbon from Glucose Is Incorporated into Wound Suberin

Following multiple bolus applications of ^13^C glucose, a subset of tissue from each labelling event was left to wound heal and collected 168 hpw. Overall, ^13^C label was found in all assessed suberin monomers, of both phenolic and aliphatic origin, in wound suberin from all labelling events (Figure 10). This confirmed that metabolites synthesized from glucose were incorporated into suberin. Total poly(phenolics) was estimated from NBO yields and represent those in the primary cell wall. Glucose-derived carbon was incorporated into phenolic monomers, and ultimately phenolic suberin, from the onset of wounding. Isotopic label was differentially allocated to poly(phenolic) monomers across the first 72 h of wound-healing with the greatest carbon incorporation from when label was applied at 48 hpw (one-way ANOVA; 0 hpw F_3,37_ = 4.8, *p* = 0.0061). Isotopic enrichment of selected representative aliphatic suberin monomers was estimated after depolymerization of extractive free tissue with MeOH-HCl (Figure 10). The composition of the PAD of potato was dominated by 18:1 dioic acid, the most abundant fatty acid derivative in aliphatic suberin, and 18:1 w-hydroxy fatty acid, the second most abundant fatty acid derivative in aliphatic suberin (Figure 11). Other monomers include very long chain fatty acids, 1-alkanols, and VLC w-hydroxy fatty acids, and dioic acids. Overall, carbon from [^13^C_6_]-glucose was incorporated into all four-representative aliphatic suberin monomers, regardless of when the label was applied. However, the proportion of label differed between monomers over time. Specifically, carbon enrichment was uniformly distributed within C22:0 1-alkanols (one-way ANOVA; 0 hpw F_3,50_ = 2.4, *p* = 0.0798) across all labelling times (Figure 10), suggesting early establishment of steady state biosynthesis, notwithstanding its relatively low abundance in potato aliphatic suberin. Although C22:0 fatty acid ^13^C enrichment appeared to lag behind that of the C22:0 1-alkanol, it appeared to be in steady-state biosynthesis early-on. Specifically, C22:0 fatty acid was consistently enriched with ^13^C from labelling events at 24, 48, and 72 hpw (one-way ANOVA; 0 hpw F_3,50_ = 6.3, *p* = 0.001). By contrast, isotopic enrichment of 18:1 dioic acid (one-way ANOVA; 0 hpw F_3,53_ = 10, *p* < 0.0001) and 18:1 ω-hydroxy fatty acid (one-way ANOVA; 0 hpw F_3,48_ = 11, *p* < 0.0001) did not reach the maximum level until 72 hpw (Figure 10). Similarly, esterified ferulic acid isotopic label enrichment increased across timepoints (Figure 10, ANOVA; 0 hpw F_3,53_ = 3.3, *p* = 0.0277). This suggested a delay in ramping up biosynthesis of the major aliphatic suberin monomers and esterified phenolic monomers, relative to that of more minor aliphatic monomers. Pairwise comparisons among suberin monomers, presented for each sampling timepoint for proportion of the pool with ^13^C label, support these conclusions Appendix A. The consistency in incorporation of label across all suberin monomers, regardless of metabolic origin, may be due to residual label from either the glucose pool and/or sucrose pool continuing to feed into metabolism over the remainder of the 168 h. Simply put, the final suberin polymer derived from the 0 h labelling event would be enriched in label derived from the glucose pool for the entirety of the time course, whereas that derived from the 72 h labelling event would not have any ^13^C incorporation from monomers synthesized prior to 72 hpw. Overall, quantifiable, and statistically relevant amounts of ^13^C label were identified in all suberin monomers based on one-sample *t*-tests in which the proportion of ^13^C label incorporation was tested against a theoretical mean of 0 Appendix A.

## 3. Discussion

Suberin is a major metabolic sink, and suberization is a highly regulated, temporal process (both transcriptionally and metabolically) [reviewed in [42]]. Herein, we performed ^13^C isotopic labelling experiments, based on [^13^C_6_]-glucose, in a wound-healing potato tuber system that produces large amounts of suberin. This allowed for time-wise assessment of carbon incorporation into suberin precursors through primary metabolism and the dynamic of carbon incorporation into the suberin polymer to be assessed. First, we establish that [^13^C_6_]-glucose was taken-up by the tissue and, second, that the supplied label was utilized for wound-induced metabolism. To address the incorporation of isotopic glucose, the freshly wounded tubers were vacuum infiltrated with a single application of 100 mM [^13^C_6_]-glucose solution, washed in sterile water, and extracted immediately following application (<0.5 h) and at 24, 48, and 72 h after application. Approximately 40% of the pre-existing glucose pool was isotopically enriched and subsequently rapidly metabolized. Isotopic enrichment of various downstream metabolites confirmed that the exogenous isotopic label was used to fuel wound-induced metabolism. During the first 72 h post label application, citric acid, shikimic acid, and L-phe rapidly approached isotopic steady state. The most label was present in these proxies of suberin monomers at 24 hpw and was consistent both proportionately and in the amount of label relative to metabolite pool up to 72 h. When [^13^C_6_]-glucose was applied to suberizing tubers at different timepoints post wounding in a multi-labelling experiment, suberin monomers were directly assessed, confirming that exogenous glucose was not only incorporated into primary metabolites but was also the source of carbon for the suberin polymer.

Working in a complex metabolic system and using an isotopic label that is the sole source of carbon to all downstream metabolites presented challenges in the derivation of the data. Namely, labelled [^13^C_6_]-glucose and the pre-existing unlabelled (predominately ^12^C) glucose pool both contribute C to downstream metabolites. In the case where there is a synthesis reaction, we can expect the product to have unique contributions that reflect the abundance of isotopic label from the substrates. In brief, the synthesis of molecules derived from shikimic acid pathway were assessed based on their synthesis from erythrose 4-phosphate (four carbon skeleton) and PEP (three carbon skeleton), whereas stearic and palmitic acids arise from acetate (two carbon skeleton) derived from pyruvate. Since only 40% or less of glucose was [^13^C_6_], a dilution of the label resulted in possibility compounds being differentially enriched. For example, any combination of 1–16/18 carbons for palmitic acid and stearic acid, respectively, could be labelled. When possible, parent ions were used for quantification and isotopic ions integrated based on predicted labelling patters based on known metabolic origins. In the case of fatty acids, detector sensitivity was a limitation and so proportion of compound containing ^13^C was calculated based on a +1, +2, +3, and +4 expected labelling pattern. More simply, the end-products (i.e., suberin phenolic and aliphatic monomers) exhibited the same labelling patterners as their upstream proxies. For example, the aromatic ring of L-phe contributed directly to phenolic monomers further confirming the precursor–product relationship [43]. A further complication was that sucrose was readily formed from [^13^C_6_]-glucose and thus represented a significant pool of isotopic carbon available for incorporation into metabolism long after the initial isotopic bolus from glucose was depleted. Because there is no way to circumvent the natural tendency of plants to sequester excess carbon as sucrose, our results were interpreted with caution and the understanding that suberin monomers derived from [^13^C_6_]-glucose applied at 0, 24, 48, and 72 hpw also encompassed carbon label released from sequestered label throughout the 168 h wound healing timeframe. In other words, the ^13^C-labelled sucrose represented an available pool of isotopic carbon that could result in more homogenous labelling of suberin monomers over time.

Previous studies have illustrated the differential induction of phenolic and aliphatic pathways and the subsequent assembly of suberin phenolics and suberin aliphatics. However, evidence has been limited to targeted and/or untargeted metabolic analyses of time course data from polar and non-polar extracts [31], histochemical data [22], and transcriptomic data [15]. While flux analyses have been applied to wound-healing tissues to delineate processes of fatty acid metabolism [44], they did not include analysis of phenolics. In the past, ^13^C-stable isotope labelling has been limited to the incorporation of products of phenylpropanoid metabolism into the suberin polymer [7] and suberin more broadly [45] while ^2^H-labelled L-phe has been used to monitor wound-induced flux through phenylpropanoid metabolism [3]. However, these authors did not encompass time-wise observations of the partitioning of carbon between branches of suberin metabolism. The present work complements these earlier studies by providing evidence that the shift between in polar and non-polar metabolism [31] and transcript accumulation of associated metabolic enzymes [15] is reflected in preferential labelling of phenolic suberin proxies early in the wound healing process. Our data, from both the single- and multi- bolus labelling experiments, in tandem with metabolite abundance data clearly demonstrate the following:Shikimic acid, an upstream precursor to L-phe, is rapidly and constantly synthesized post wounding, thus priming the system for feeding downstream metabolites; notably, shikimic acid accumulates in wound healing tissue indicating that source > sink. At the same time, L-phe is rapidly depleted, while also containing significant ^13^C labelling, indicating that although it is being readily made, source < sink. Conversely, citric acid, which represents a large supply of label, was rapidly enriched with ^13^C label immediately post wounding. The citric acid pool, which is exported to the cytoplasm and cleaved to form acetate that feeds the formation of VLCFAs, is stable up until 48 hpw and then depleted by 72 hpw. This shift may represent achieving steady state early-on, where carbon supply to the TCA cycle is fueling baseline cellular functions before a shift 72 hpw where there is increased demand for citric acid, i.e., source < sink. This depletion of the citric acid pool coincides with the timepoint where the most label is incorporated into 18:1 w-hydroxy fatty acid and 18:1 dioic acid. Lastly, it is difficult to assess if there is a substrate–product dynamic between palmitic and stearic acids and downstream aliphatic suberin monomers. Acetate liberated from pyruvate, neither of which were identified in our extracts, is responsible for production of palmitic and stearic acids. Despite the availability of label in intermediates of glycolysis, evident from label found in the TCA cycle and shikimic acid (derived from pyruvate), little label was found in palmitic or stearic acids within the first 72 hpw.The dynamic of carbon turnover in L-phe was different than that of stearic and palmitic acids. Differences in the total amount of pre-existing metabolites in the tissue may result in higher labelling proportions in less abundant metabolites. It is possible that the greater proportion of label found in L-phe, in comparison to palmitic and stearic acids, could be due to a dilution of label as it entered the larger pre-existing pools of palmitic and stearic acids. Therefore, one cannot rely solely on differences in proportional enrichment of the metabolites to gauge the relative importance of one molecule’s biosynthesis over another’s. Additionally, it is possible that the high proportion of label in L-phe may be an outcome of a slow enzymatic turnover by the phenolic entry point enzyme PAL [46,47]. However, consistent incorporation of relatively high proportionate amounts of isotopic label into L-phe across 72 h strongly indicates active synthesis, and not a slow turnover. Furthermore, the total amount of L-phe in the tissue does not accumulate during the first 72 hpw and thus must be consumed at least as rapidly as it is made. In the single [^13^C_6_]-glucose application experiment, where total L-phe decreased following wounding, a strong argument can be made that the rate of utilization of L-phe was greater than that of its synthesis. Conversely, stearic and palmitic acids had proportionately less label than L-phe, while both pools showed little change overtime, implying that their pools were more static.Carbon from [^13^C_6_]-glucose was consistently allocated towards phenolic monomers at all timepoints, with the greatest being from the 48 hpw labelling event. The efficient labelling of L-phe across all timepoints supports the degree of label found in the poly(phenolics) regardless of when label was applied. Overtime, there was increasing dedication of carbon towards aliphatic monomers, such as 18:1 ω-hydroxy fatty acids and 18:1 dioic acids, which comprise the bulk of aliphatic suberin in potato. More carbon was dedicated to these compounds after label application at 72 hpw than from either 0 or 24 hpw applications of [^13^C_6_]-glucose. Unlike that of the phenolic proxies, there was no indication of significant turnover of aliphatic proxies in the first 72 hpw, so it is likely that contribution of ^13^C-carbon occurred later than 72 hpw. This temporal pattern is also exhibited in transcriptomic data wherein accumulation of transcripts involved in fatty acid biosynthesis were delayed relative to those involved in phenolic biosynthesis [16], reflective of potential differential regulation by ABA [37]. Moreover, the level of membrane fatty acids (predominantly 16:0, 18:2 and 18:3 in potato tubers [43]) remained constant through out wound healing, but especially during the early dpw, while suberin-associated fatty acids only began to accumulate after 3–4 dpw [44]. Alternatively, it is possible that suberin monomers may derive from the stearic acid and palmitic acid that exists in the tissue prior to wounding, which may explain why less label is invested into aliphatics from early timepoints. However, when metabolite abundance was tracked over the 168 h time course, palmitic acid and stearic acid gradually increased in the tissue without substantial ^13^C-label, making pre-existing compounds an unlikely source for incorporation into suberin. Lastly, it is also possible that labelled carbon from [^13^C_6_]-glucose was less efficiently incorporated into new fatty acids, at least at a time when much of the applied label was rapidly channelled through phenolic metabolism. For example, there was less ^13^C available for incorporation into the suberin polymer due to lesser proportion of label at the 0 h labelling event from the multi-labelling trial and this may be reflected in less proportional enrichment of the suberin polymer from label applied at 0 hpw. A comparison between 24, 48, and 72 hpw labelling events is, therefore, more valid, as similar proportions of [^13^C_6_]-glucose were available, and glucose was depleted at similar rates.Lastly, the incorporation of ^13^C-label into esterified ferulic acid mirrored the labelling pattern of 18:1 w-hydroxy fatty acid and 18:1 dioic acid, and not that of the poly(phenolic) monomers. Superficially, C22:0 fatty acid and C22:0 1-alkanol also show similar labelling patterns to phenolic monomers; however, these compounds are not very abundant in suberized tissue as observed in the chromatogram (Figure 11), and thus the outcome may be due to computational limitations of integrating isotopic ions that are nearing the limit of detection of the instrument.

At the metabolic level, our results demonstrate a dynamic picture of suberin metabolism during wound-induced suberization in support of previous results showing a shift between from polar and non-polar metabolism during suberization. Specifically, our analysis has provided evidence of the dynamic of carbon sharing between phenolic- and aliphatic-destined metabolic pathways. We conclude that carbon turnover through phenylpropanoid metabolism is prioritized over that of fatty acid metabolism during the early stages of wound-induced suberization. At the structural level, the data from the consecutive labelling approach (where ^13^C label was applied, 0, 24, 48, and 72 hpw) supports the hypothesis that phenolics are first polymerized in the primary cell wall followed by the polymerization of aliphatic monomers [reviewed in [6,8,15]]. Thus, our data supports a temporal deposition model where the PPD acts as an anchor and foundation to which the aliphatic lamellae can attach. This is consistent with the observation that when the PPD is not formed properly in Arabidopsis endodermal cells, the aliphatic suberin lemellae are not deposited [27]. The temporal difference between phenolic monomer and aliphatic monomer biosynthesis is expected to influence the availability of suberin monomers for assembly, thus these data may corroborate that assembly into spatially distinct domains is regulated first at a biosynthetic level. At a functional level, it was also previously established that both polymers must be properly formed to confer resistance to pathogens [22] and dehydration [23,25,26]. In a single cell, it can be surmised that resistance is gradual. First, a partial barrier is formed as the PPD is synthesized and assembled and later the barrier is completed as the PAD is deposited. The presence of both a PAD and PPD in the same tissue, dynamics of metabolic regulation of the branches of suberin biosynthesis, proposed reliance of the PAD on the PPD, and the involvement of both domains to achieve a functioning barrier poses that both polymers must be considered when assessing suberized tissues.

## 4. Materials and Methods

### 4.1. Biological Material

Potato tubers (*Solanum tuberosum* L. cv. Russet Burbank) were collected from plants grown at the Environmental Science Western Field Station, Ilderton, ON, Canada, under standard field conditions. Tubers were stored in the dark at 5 °C until use.

### 4.2. Chemicals and Reagents

All chemicals and reagents were purchased from Sigma-Aldrich unless specified otherwise. Solvents were HPLC-grade. The [^13^C_6_]-glucose (Sigma-Alderich Cat: 389374, St. Louis, MO, USA) was ≥99% atom % ^13^C. 

### 4.3. Study Design

To establish the temporal nature of carbon allocation across wound-induced metabolism, we adopted two approaches: (1) a single bolus of [^13^C_6_]-glucose administered immediately post wounding with the tissue collected at 0 (i.e., the time of application), 24, 48, 72, and 168 hpw, with unlabelled (control) tissue collected in tandem; (2) multiple bolus applications of [^13^C_6_]-glucose infiltrated into unlabelled tissue at 24 h intervals for the first 3 days of wound healing;, i.e., at 0, 24, 48, 72 hpw. The [^13^C_6_]-glucose-treated tissues were sampled 0, 1, 3, and 6 h post labelling, providing ‘snapshots’ of C-partitioning every 24 hpw. The unlabelled (control) tissues were sampled every 12 h for the first 4 days post wounding, beginning immediately after wounding, and then at 24 h intervals for the remainder of the 7-day (168 h) time course. The single bolus labelling experiment was conducted once (n = 5/timepoint), while the consecutive labelling experiment was conducted 3 times with (n = 5/timepoint); individual tubers were treated as one replicate, with discs from the same tuber used across all sampling timepoints.

### 4.4. Wounding and Label Infiltration

Tuber wounding and collection of suberizing tissue was as described in [16], with minor modification. Briefly, potato tubers were cleaned under running water, surface sterilized in a 1.25% (*v*/*v*) sodium hypochlorite solution for 1 h, and air-dried in a laminar flow hood. The tubers were then sectioned into 1 cm thick slices with individual discs excised using a 1 cm diameter cork borer. Discs were placed in a sterile water bath to delay wound-healing processes. Discs derived from each replicate tuber were placed in separate sterile Magenta^®^ boxes (Sigma-Alderich, St. Louis, MO, USA) (9 discs per box) fitted with wet filter paper under a stainless-steel mesh platform and incubated at 25 °C for up to 7 days (five replicates per treatment), with the suberizing surface layers collected at the previously described intervals. For infiltration of the [^13^C_6_]-glucose solution, a manual vacuum infiltration method was used: briefly, 12 discs were stacked into a sterile 20 mL syringe with 5 mL of 100 mM [^13^C_6_]-glucose. Next, negative pressure was generated inside the syringe barrel by plugging the outlet and pulling the plunger, this was performed repeatedly for approximately 5 min. Tissue discs were then rinsed with sterile water, blotted with a sterile KimWipe^®^ (Kimberly-Clarke, Irving, TX, USA), and placed back into their sterile chambers to continue wound healing.

### 4.5. Bi-Phasic Extraction

Soluble metabolites were extracted using MeOH–methyl–*tert*–butyl ether–water (1:3:1), as adapted from Giavalisco et al. (2011) [48]. In brief, 50 mg frozen tissue was first suspended in 500 µL of cold (−20 °C) 50% *v*/*v* MeOH (containing 0.02 mg/mL ribitol as internal standard) before 750 µL of cold (−20 °C) methyl–*tert*–butyl ether (MTBE) was added. The samples are vortexed, placed on a rotating mixer in the cold room (4 °C) for 30 min, moved to a sonicating bath for 15 min, and centrifuged (14,000× *g*) for 10 min. Three 100 µL aliquots of the lower, polar (i.e., MeOH-H_2_O) phase were transferred to separate clean microcentrifuge tubes, dried under a stream of N_2_, and stored at −20 °C until further analysis.

### 4.6. Primary Metabolite Analysis

Aliquots of the dried polar extracts were derivatized using a two-step process involving methoximation followed by trimenthylsilylation, using a protocol adapted from Fiehn (2016) [49]. Briefly, 20 µL of 20 mg/mL methoxyamine–HCL in pyridine was added to the dried extract residues and the samples incubated at 30 °C for 90 min. Next, 80 µL of methyl-N-trimethylsilyl trifluoroacetamide (MSTFA) was added to each sample and incubated for 30 min at 37 °C. Derivatized samples were cooled to room temperature, centrifuged, and 10 µL was transferred to a chromatography vial fitted with glass micro-volume inserts and diluted with chloroform to a final volume of 100 µL. Samples were immediately analyzed via GC-MS on an Agilent 7890A GC coupled with a LECO Pegasus BT time of flight MS. Liquid samples (1 µL) were injected in splitless mode onto a RESTEK (Restek Corp., Bellefonte, PA, USA) Rxi-5ms low bleed GC column (30 m, 250 µm internal diameter and 0.25 µm film thickness) (Restek: cat. 709-809-508) and eluted with the following oven temperature programme: initial temperature at 50 °C held for 0.5 min followed by a temperature ramp at 20 °C min^−1^ to a final temperature of 325 °C and held at 325 °C for 5.75 min. Injector and transfer line temperatures were set to 275 °C. The ion source was set at 250 °C. High-purity helium was used as carrier gas at a flow rate of 1 mL min^−1^. Data were acquired over a 50–500 *m*/*z* range (17 spectra/s) after a 3 min delay to allow the solvent to clear the system. Primary metabolic analysis samples were run in cohorts grouped by collection timepoint to reflect the experimental design. Each cohort included an injection of chloroform, 6-point quality control mix based on Fiehn (2016) [49], retention index mix, and reagent blank at the start of each cohort. For calibration, a dilution series was prepared in triplicate for target analytes (10 µg/mL–0.03125 µg/mL) and derivatized and analyzed as above.

### 4.7. Aliphatic Suberin Analysis

For the analysis of aliphatic suberin monomers and suberin-associated waxes ~200 mg of finely ground, frozen, day 7 wound-periderm was Soxhlet extracted, derivatized and analyzed by GC-MS essentially as described by Meyer et al. (2011) [50]. Briefly, tissue was extracted in a Soxhlet apparatus twice for 3.5 h with 50 mL CHCl_3_/CH_3_OH (2:1; *v*/*v*) and overnight (12 h) with 80 mL of CHCl_3_. Following Soxhlet extraction the extractive-free residue was washed with acetone and air dried. The dried insoluble residue (~10 mg) was trans-esterified in 500 µL of 3 M HCl in methanol at 80 °C for 2 h. An equal volume of sodium chloride saturated water was added to stop the reaction along with 5 µL of 1 mg/mL triacontane internal standard. The aqueous phase was extracted 3 times with 1 mL of n-hexanes, with the extracts pooled and dried under a stream of N_2_. The recovered aliphatics were TMS-derivatized with 25 µL each of pyridine and BSTFA at 70 °C for 40 min. Samples (1 µL) were injected onto a RESTEK Rxi-5ms low bleed GC column (30 m, 250 µm internal diameter and 0.25 µm film thickness) (Restek: Cat. 709-809-508) in splitless mode and eluted with the following oven temperature programme: initial temperature at 70 °C held for 2 min followed by an initial temperature ramp (40 °C min^−1^) to 200 °C, held for 2 min and a second temperature ramp (3 °C min^−1^) to a final temperature of 320 °C and held for 9.4 min. Data acquisition was turned on after a 4 min delay to allow the solvent to clear the system. Injector and transfer line temperatures were set to 275 °C. The ion source was set at 250 °C. High-purity helium was used as carrier gas at a flow rate of 1.2 mL min^−1^. Data were acquired over a 50–500 *m*/*z* range (17 spectra/s) after a 6 min delay to allow the solvent to clear the system.

### 4.8. Micro-Scale Nitrobenzene Oxidation (NBO)

Micro-scale nitrobenzene oxidation (NBO) [51] was conducted, with modifications in addition to those described by Thomas et al. (2007) [52], to quantify and assess the composition of the phenolic portion of suberin. Dried extractive-free periderm tissues (~20 mg) were saponified with 5 mL of 1 M NaOH for 24 h at 37 °C. To recover the lipid-free residue, the tissue was pelleted at 5600 G for 10 min and supernatant discarded. The pellet was washed with 5 mL ultrapure water (3 times), 5 mL 80% ethanol, and 5 mL 100% acetone with the tissue pelleted and kept by centrifugation each time. The dried, saponified residue was stored at 4 °C. In 2 mL glass ampoules, ~1.7 mg of dried residue was suspended in 300 µL of 2 M sodium hydroxide and 15 µL of nitrobenzene. Alternatively, ~1.7 mg of dried reside was suspended in 300 µL of 2 M sodium hydroxide and 15 µL of nitrobenzene in 5 mL glass ampoules. The ampoules were flame-sealed and incubated at 160 °C for 3 h. Samples were cooled to room temperature, internal standard added (5 µL of 1 mg/mL 3 ethoxy-4-hydroxybenzaldehyde), and the samples quantitatively transferred to 4 mL glass vials with sodium chloride saturated water. The mixture was extracted thrice with dichloromethane (1 mL), and the aqueous phase acidified with 6 M hydrochloric acid (HCl) (Thermo-Fisher Scientific, Waltham, MA, USA). Samples were then extracted three times with 900 µL of hexanes and the hexane phases were pooled, dried with anhydrous sodium sulphate, and evaporated under a stream of N_2_. Dried residues were TMS-derivatized with 25 µL each of pyridine and BSTFA at 70 °C for 40 min. Phenolic derivatives produced from nitrobenzene oxidation were analyzed by GC-MS using the elution parameters described for the polar primary metabolite analysis. For quality assurance, a potato native periderm preparation, which has a known NBO profile, and a mix containing the internal standard and all six main NBO phenolic products were also analyzed.

### 4.9. Data Analysis

#### 4.9.1. GCMS Data Processing

For metabolite abundance measurements of GC-MS data, data were baseline corrected, deconvoluted and aligned using ChromaTOF-Sync (Leco Corp., St. Joseph, MI, USA). Analyte annotation by mass spectra features was based on the NIST main library. Data cleaning was performed on the normalized, aligned data matrix obtained from ChromaTOF-Sync^®^. Only components with peak areas greater than the average peak area of five reagent blanks were retained and imported into MetaboAnalyst 5.0 (https://www.metaboanalyst.ca; accessed on 10–17 February 2025). To assess the overall precision of the analytical method, groups of composite samples were plotted on a PCA plot to assess total variance allowing for specific identification of batch defects. In addition, to control for efficiency of derivatization, a comparative check on the 6 pt—quality control mix was performed with each batch as detailed in Fiehn (2016) [49]. Parameters for One Factor—Statistical Analysis in MetaboAnalayst were selected as follows: Data filtering was performed using standard deviation (RSD = SD/mean) at 30% and normalization was performed using ribitol as a reference feature. Lastly, the matrix was assessed for normal distribution and data were square root transformed and scaled with Pareto scaling (mean-centred and divided by the square root of the standard deviation of each variable).

#### 4.9.2. Calculation of the Proportion of Compound with Label

For proportional isotope measurements, mass shifts were predicted for representative mass fragments and peak areas integrated using ChromaTOF (Leco Corp.) software (V5.51.06). To calculate the proportion of molecules containing ^13^C, *PoM*^13^*C*, the sum of the integrated peak area for all predicted isotope-enriched fragments ($\Sigma {Peak\;Area}_{Heavy})$ for a given natural abundance fragment (${Peak\;Area}_{Light})$ were used:PoMC13=ΣPeak AreaHeavyPeak AreaLight+ΣPeak AreaHeavy

Lastly, the *PoM*^13^*C* of compounds derived from the unlabelled (control) tissue was calculated and used to correct the *PoM*^13^*C* of the labelled compounds. Labelled fragments were predicted by considering the biosynthetic pathway from glucose to the molecule of interest, and how many C’s in each fragment could come from ^13^C-glucose. For example, L-phe is derived from two molecules of P-*enol*-pyruvate and one erythrose-4-P via the shikimic acid pathway. One major fragment of the L-phe 2TMS derivative, at *m*/*z* 192.12, derives from the loss of CO_2_Si (CH_3_)_3_ leaving C_11_H_18_NSi. However, only 8 Cs derive from L-phe, of which x, y, or z could be ^13^C labelled, depending on whether one, two, or all three precursor metabolites derive from ^13^C-glucose. In this case, we summed the +x, +y and +z peak areas in the labelled sample, corrected for the same *m*/*z* from the unlabelled sample to calculate the *PoM*^13^*C*. Lastly, the total amount of metabolite in the tissue was multiplied by the *PoM*^13^*C* to obtain a measure of amount of ^13^C-enriched compound (µg/mg dry tissue).

To calculate the proportion of total phenolics with ^13^C label, the proportion of ^13^C label in the products of nitrobenzene oxidation were quantified for individual samples. Peak areas of each NBO product were normalized across samples and were quantified using calibration standards (10–0.0625 µg/mL). The total amounts (µg/mg of tissue) of each compound per sample were then multiplied by PoM^13^C of the same sample. Total amounts of compound containing label were then summed and divided by the sum of the NBO products.

#### 4.9.3. Statistical Analysis

All statistical analyses were performed in Prism (V 9.5.1). Metabolite abundance and *PoM*^13^*C* data were either analyzed by one-way ANOVA followed by Tukey’s multiple comparisons test or a one-sample *T*-Test. Significance cut-off was set at *p* = 0.05. Depletion of ^13^C label in glucose over 6 h post administration was assessed by simple linear regression.

## Figures and Tables

**Figure 1 plants-14-01433-f001:**
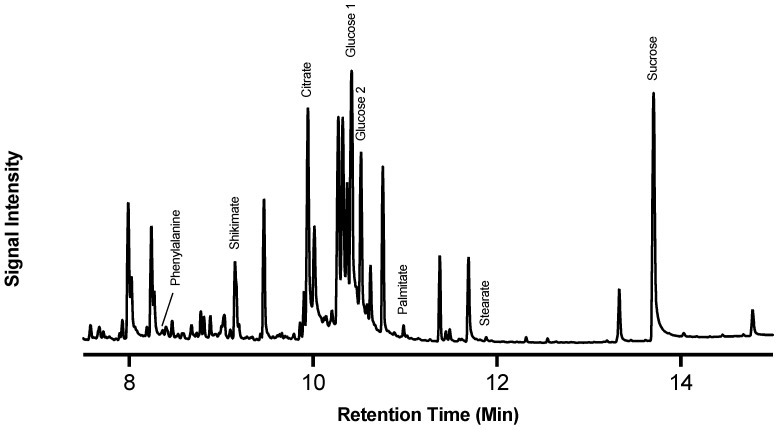
Profile of potato polar metabolites. Total ion chromatogram (TIC) from gas chromatography–mass spectrometry of polar metabolites. Data presented are representative of the polar TIC profile in freshly wounded potato tissue. Proxy metabolites were identified in the TIC using retention time, mass spectra, and true standards. The most abundant metabolites are glucose and sucrose. IS = adonitol internal standard.

**Figure 2 plants-14-01433-f002:**
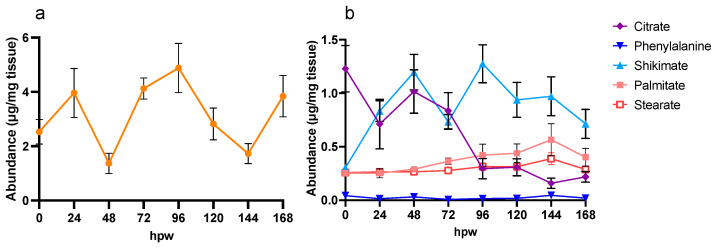
Targeted metabolic analysis of proxy metabolites selected for their role in suberin monomer biosynthesis. Proxy metabolites were identified and measured over 7 days of wound healing. (**a**) Sucrose (yellow circles) is presented independently from (**b**) all other proxies as it is more than twice as abundant in tissue. Metabolite abundance is represented by quantified amount in µg/mg of dry tissue. Data points represent mean ± SEM (n < 15). hpw = hours post wounding.

**Figure 3 plants-14-01433-f003:**
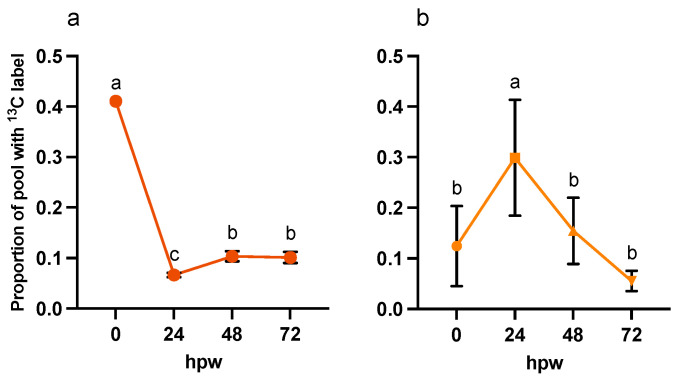
Confirmation of uptake and metabolism of (**a**) [^13^C_6_]-glucose following a single bolus application of [^13^C_6_]-glucose and subsequent label quantified in (**b**) sucrose. Label was applied at the time of wounding and the tissue was sampled every 24 h for 72 h. The data represent the proportion of the metabolite pool enriched with ^13^C label. Data were analyzed by one-way ANOVA, followed by Tukey’s multiple comparisons test. Different letters above bars indicate significant differences (*p* < 0.05). Data points represent the mean values normalized to the average proportion of glucose with ^13^C label at the time of labelling (0 h post label application) ± SEM (n = 5). hpw = hours post wounding.

**Figure 4 plants-14-01433-f004:**
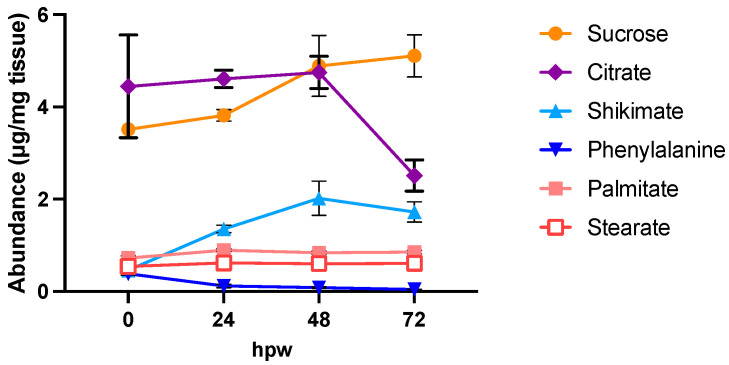
Proxy metabolite abundance in suberizing tissue treated with a bolus of [^13^C_6_]-glucose at the time of wounding. Proxy metabolites were identified and measured over 72 h of wound healing after treatment with 100 mM ^13^C_6_-glucose. Metabolite abundance is represented by quantified amount in µg/mg of dry tissue. Data points represent mean ± SEM (n = 5). hpw = hours post wounding.

**Figure 5 plants-14-01433-f005:**
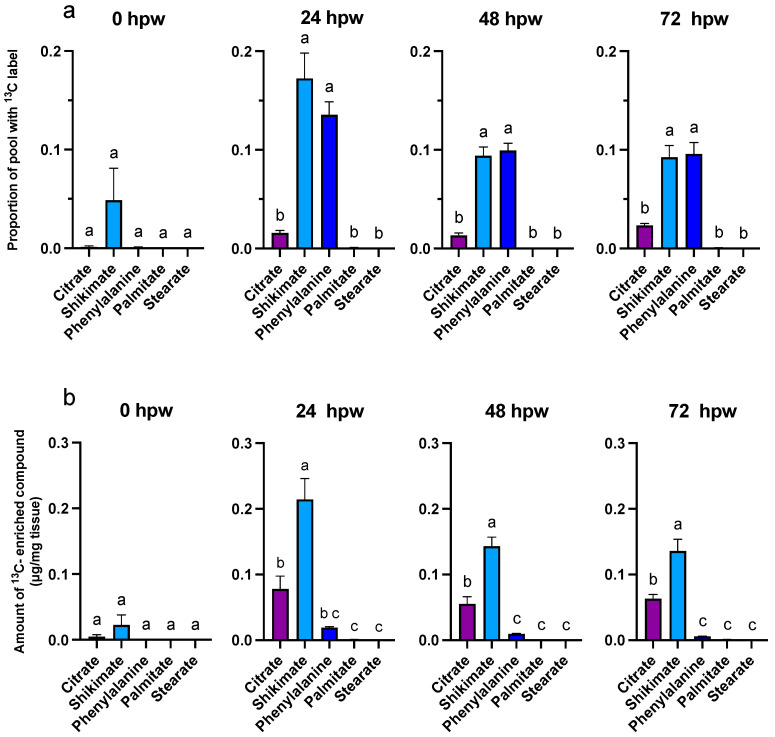
Metabolite pool enrichment with ^13^C label following a single bolus application of [^13^C_6_]-glucose. Label was applied at the time of wounding and the tissue was sampled at 24, 48, and 72 hpw. (**a**) The proportion of ^13^C-enriched compound relative to the (**b**) total amount of ^13^C-enriched metabolite present in the tissue (ug/mg tissue) are presented. ^13^C-enriched metabolite values were calculated by multiplying the proportion of metabolite pool with ^13^C label by the total amount metabolite present in the tissue at the time of sampling (See Figure 3). Replicates were normalized to the average proportion of ^13^C glucose in the tissue at 0 hpw. Data were analyzed by one-way ANOVA, followed by Tukey’s multiple comparisons test. Different letters above bars indicate significant differences (*p* < 0.05). Bars represent mean ± SEM (n = 5). hpw = hours post wounding.

**Figure 6 plants-14-01433-f006:**
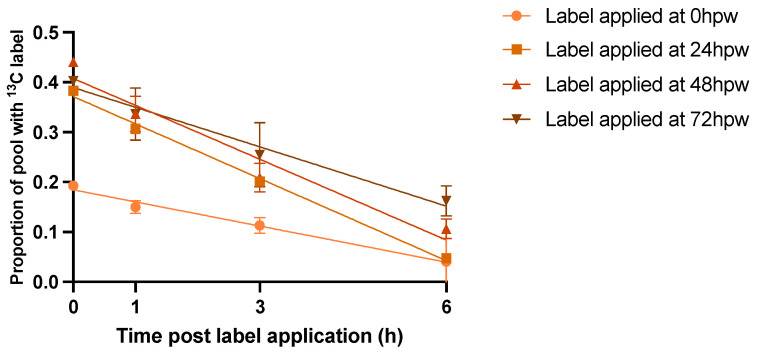
Multiple applications of [^13^C_6_]-glucose to wound healing potatoes. [^13^C_6_]-glucose was applied to tissue at the time of wounding and un-labelled wound-healing tissue 24, 48, and 72 hpw and tissue was sampled immediately following label application, 1, 3, and 6 h later. Data points represent the proportion of the metabolite pool enriched with ^13^C label. Lines represent the linear regression line of best fit Appendix A. Replicates were normalized to the average proportion of ^13^C glucose in the tissue at 0 h post label application for each of the time courses. Data were analyzed by simple linear regression. Data points represent mean ± SEM (n = 10–15).

**Figure 8 plants-14-01433-f008:**
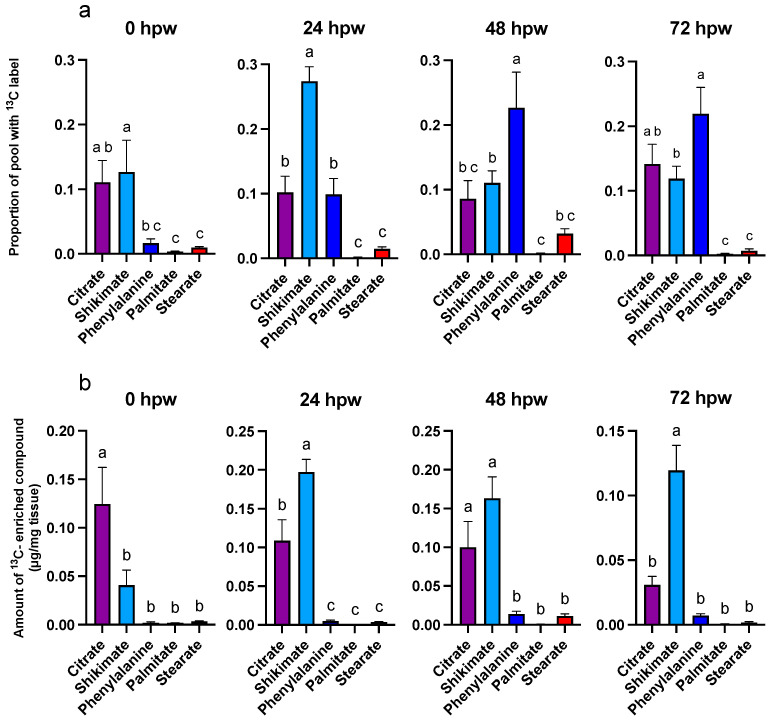
Metabolite pool enrichment with ^13^C label following independent applications of ^13^C_6_-glucose at different times post wounding. ^13^C_6_-glucose was applied to tissue at the time of wounding and to unlabelled wound-healing tissue 24, 48, and 72 hpw, and the tissue sampled 3 h later. (**a**) The proportion of ^13^C-enriched compound and (**b**) the amount of ^13^C-enriched compound, relative to the total amount of ^13^C-enriched metabolite present in the tissue (µg/mg tissue), are presented. ^13^C-enriched metabolite values were calculated by multiplying the proportion of metabolite pool with ^13^C label by the total amount metabolite present in the tissue at the time of sampling (See Figure 9). Replicates were normalized to the average proportion of ^13^C glucose in the tissue at 0 h post label application for each of the time courses. Data were analyzed by one-way ANOVA, followed by Tukey’s multiple comparisons test. Different letters above bars indicate significant differences (*p* < 0.05). Data represent the mean normalized values mean ± SEM (n = 9–15).

**Figure 9 plants-14-01433-f009:**
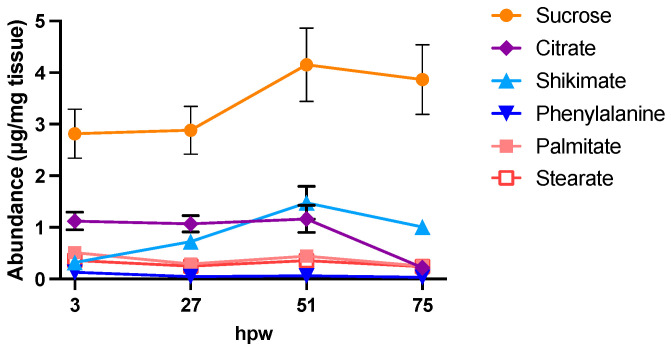
Proxy metabolite abundance in suberizing tissue treated with independent applications of [^13^C_6_]-glucose at different times post wounding. Data points represent the abundance of proxy metabolites 3 h post label application, where label was applied to subsets of unlabelled wound-healing tissue at 0, 24, 48, and 72 hpw. The time when tissue was sampled, relative to the initial wounding event, is shown on the x-axis. Metabolite abundance is represented by quantified amount in µg/mg of dry tissue. Data points represent mean ± SEM (n = 10–15). hpw = hours post wounding.

**Figure 10 plants-14-01433-f010:**
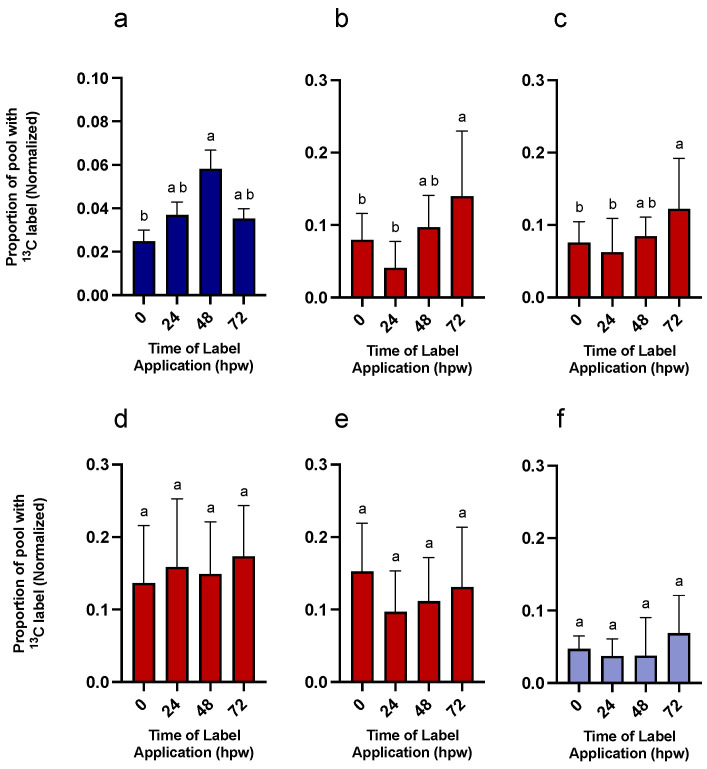
End point analysis of suberin-derived phenolic and aliphatic monomers from tissue labelled with independent applications of ^13^C_6_-glucose during early wound healing. ^13^C_6_-glucose was applied at 0, 24, 48, and 72 h post-wounding and sampled 168 hpw. Replicates from each time of label application, were normalized to the average proportion of ^13^C label found in glucose at the time of labelling (**a**) Six nitrobenzene oxidation products were summed to give a measure of the label incorporation per total phenolics. Aliphatic monomers (**b**) 18:1 dioic acid, (**c**) 18:1 w-hydroxy fatty acid, (**d**) C22:0 fatty acid, (**e**) C22:0 1-akanol, and (**f**) esterified ferulic acid were released during depolymerization of the organic solvent–insoluble residue. Data were analyzed by one-way ANOVA, followed by Tukey’s multiple comparisons test. Different letters above bars indicate significant differences (*p* < 0.05). Data points represent mean ± SEM (n = 9–15). hpw = hours post wounding.

**Figure 11 plants-14-01433-f011:**
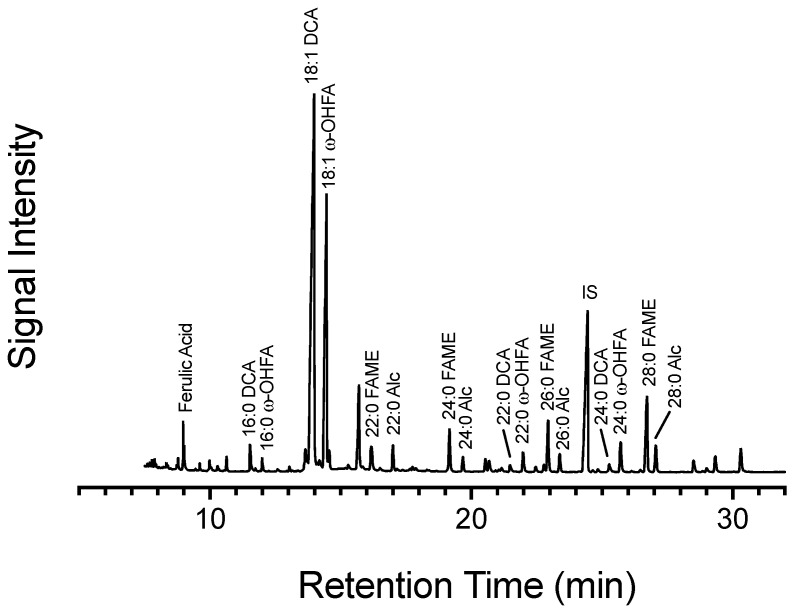
Chromatographic profile of extractive free suberin poly(aliphatics). Total ion chromatogram (TIC) from gas chromatography–mass spectrometry of the poly(aliphatic) monomers from 168 hpw potato wound periderm. Monomers of aliphatic suberin were identified in the TIC using retention time, mass spectra, and true standards. IS = triacontane, internal standard.

## Data Availability

The datasets generated during and/or analyzed during the current study are available from the corresponding author upon reasonable request.

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
