# Peer review of "Timing Is Everything: The Metabolic Partitioning of Suberin-Destined Carbon"

_plants, 2025, doi:10.3390/plants14101433_

Round 1
Reviewer 1 Report
Comments and Suggestions for Authors
In this manuscript, the authors track the incorporation of [13C6]-glucose into downstream metabolism after wound-induced suberization in potato tubers. The study further explores carbon distribution into phenolic and aliphatic metabolisms. The work is well done, and the results contribute to a better understanding of the metabolic carbon partitioning for suberin synthesis in plants.
My main criticism relates to the lack of a figure showing isotopic-marked glucose partition into the different metabolites with time in unwounded samples. To this reviewer, even many of the metabolites would be absent. A figure showing the different patterns between wounded and unwounded conditions would contribute to the clarity of the overall change in metabolism.
Minor criticisms are:
The text should be coherent in terms of the writing of time units. Better if coherently are used abbreviations h and min for hour(s) and minute(s), respectively. Also coherent should be the use of capital letters in subtitles and axis labels of figures. Otherwise, it should read "(Figures 8 & 9)" (line 340); "[Reviewed in 37]" (line 432); and "PEP" (in line 460, as the abbreviation was introduced in line 56).
Author Response
Comment 1: My main criticism relates to the lack of a figure showing isotopic-marked glucose partition into the different metabolites with time in unwounded samples. To this reviewer, even many of the metabolites would be absent. A figure showing the different patterns between wounded and unwounded conditions would contribute to the clarity of the overall change in metabolism.
Response 1: Stored tubers are largely quiescent organs, with little metabolic activity or changes to metabolites. It is only after wounding (or sprouting) that metabolism is rapidly induced. In earlier work (Woolfson et al. Phytochemistry 2023, 206, 113529. https://doi.org/10.1016/j.phytochem.2022.113529.), we demonstrated that within 12 hours of wounding over 5000 genes are differentially expressed, kick-starting the wound response. It is not possible to administer [13C6]-glu to “resting” tubers and have the label distributed across the metabolism. The best we can do is extract tuber tissue at the time of wounding, before any substantial induced metabolism has taken place. Values for glucose, sucrose, and proxy compounds are evident in Figures 2, 3, and 4, at the 0 hpw time point.
Comment 2: The text should be coherent in terms of the writing of time units. Better if coherently are used abbreviations h and min for hour(s) and minute(s), respectively.
Response 2: We have gone through the entire manuscript making time units consistent.
Comment 3: Also coherent should be the use of capital letters in subtitles and axis labels of figures.
Response 3: We have gone through the entire manuscript and made all subtitles and axis labels consistent.
Comment 4: Otherwise, it should read "(Figures 8 & 9)" (line 340); "[Reviewed in 37]" (line 432); and "PEP" (in line 460, as the abbreviation was introduced in line 56).
Response 4: these edits have been made within the text.
Reviewer 2 Report
Comments and Suggestions for Authors
In this study, the authors analyzed the temporal partitioning of carbon between phenolic and aliphatic pathways during wound-induced suberization in potato tubers. The manuscript was interesting and helpful. However, there are some questions for me to understand.
1.The rationale for selecting 0-, 24-, 48-, and 72-hour post-wounding (hpw) timepoints needs justification. Are these intervals aligned with known transcriptional or metabolic milestones in potato suberization?
2. The choice of proxies is logical, but limitations (e.g., dilution effects in large palmitic/stearic acid pools) should be explicitly addressed in the discussion.
3. Please check the significant difference in figures.
4. The discussion focuses on metabolic flux but does not fully contextualize the findings in terms of suberin’s functional roles (e.g., barrier formation, pathogen resistance). How do the observed carbon allocation dynamics influence the structural/functional properties of the suberized tissue?
Author Response
Comment 1: The rationale for selecting 0-, 24-, 48-, and 72-hour post-wounding (hpw) timepoints needs justification. Are these intervals aligned with known transcriptional or metabolic milestones in potato suberization?
Response 1: We chose 0, 24, 48 and 72 hours for our labelling timepoints for two reasons: First, these timepoints reflect when the greatest transcriptional changes induced by wounding occur, and second, they precede closing layer formation and therefore are conducive to [13C6]-glu application. We did not attempt to administer [13C6]-glu beyond 72 hpw since formation of the closing layer would impede uptake.
Comment 2. The choice of proxies is logical, but limitations (e.g., dilution effects in large palmitic/stearic acid pools) should be explicitly addressed in the discussion.
Response 2: The reviewer raises a good point; however, our data suggests that during the early stages of wound-induced metabolism there is little new fatty acid biosynthesis, and that the pool of palmitate/stearate is relatively small. This is further evidenced by the fact that the amount of palmitate and stearate remain constant, with no significant label accumulation. We have added text in reply to Comment 4 that in part addresses this comment as well.
Comment 3. Please check the significant difference in figures.
Response 3: We have checked all figures and fixed all axis labels and titles to make them consistent.
Comment 4. The discussion focuses on metabolic flux but does not fully contextualize the findings in terms of suberin’s functional roles (e.g., barrier formation, pathogen resistance). How do the observed carbon allocation dynamics influence the structural/functional properties of the suberized tissue?
Response 4: The main point of the manuscript was to establish the temporal sequence of metabolism with respect to phenolic and aliphatic suberin biosynthesis, so this is where we focussed our discussion. However, the reviewer makes a good point, and we have added some text to the introduction to (lines 71-78 in the revised manuscript) to introduce the functional properties of suberized tissue, which is followed up in the discussion to further contextualize our findings (lines 610-627 in the revised manuscript).
Reviewer 3 Report
Comments and Suggestions for Authors
This study revealed the carbon allocation dynamics of suberin metabolism during the wound - healing process of potato tubers through the stable isotope labeling technique ([13C6] - glucose tracing), and for the first time, confirmed the hypothesis that phenolic metabolism is activated prior to lipid metabolism from the perspective of metabolic flux. The paper has a clear logic, but there are the following problems:
- The references cited in the introduction section are not up - to - date, resulting in a lack of cutting - edge discussion on the structural controversies. Please supplement the research on suberin structure and regulatory mechanisms in the past five years.
- Figure 8 is repeatedly written. And for sub - figures (such as Figure 10), the specific p - value ranges for significant differences are not marked, including Figure 3, Figure 8, Figure 10, etc.
- The discussion does not clearly link the results directly with the core hypothesis proposed in the introduction ("The temporal sequence of carbon allocation is driven by transcription - metabolism coupling").
- The explanation for the delay of lipid metabolism lacks a comparison with key studies on lipid synthesis.
- In the Materials and Methods section, the key instrument settings for GC - MS are missing, such as the inlet temperature, ion source temperature, and scanning mode.
- The manufacturers of some reagents are not marked, such as "triacontane internal standard".
Author Response
Comment 1: The references cited in the introduction section are not up - to - date, resulting in a lack of cutting - edge discussion on the structural controversies. Please supplement the research on suberin structure and regulatory mechanisms in the past five years.
Response 1: The reviewer makes a good point; however, we have tried to use original references to support our arguments where possible. Much of the motivation for the work presented comes from the fact that the phenolic domain of suberin has received little attention in the past and most of the more recent papers describing structural aspects of suberin have focussed on the aliphatic domain only. Similarly, the recent emphasis on transcription factor regulation of suberin biosynthesis has also focussed on the aliphatic domain. We have clarified in the text that there is a gap in the literature regarding polar metabolism in wound-healing potato tubers and added additional text describing more recent work to the introduction (lines 109 – 116 in the revised manuscript)
Comment 2: Figure 8 is repeatedly written. And for sub - figures (such as Figure 10), the specific p - value ranges for significant differences are not marked, including Figure 3, Figure 8, Figure 10, etc.
Response 2: We have removed repetitive reference to Figure 8, and added the statement “Data were analyzed by one-way ANOVA, followed by Tukey’s multiple comparisons test. Different letters above bars indicate significant differences (p < 0.05).” to the three figures lacking this information. In addition, we have added a new section 4.8.3 “Statistical Analysis” describing the statistical tests used to analyze our data.
Comment 3: The discussion does not clearly link the results directly with the core hypothesis proposed in the introduction ("The temporal sequence of carbon allocation is driven by transcription - metabolism coupling").
Response 3: We have revised the discussion to address this comment (lines 614-632 in the revised manuscript).
Comment 4: The explanation for the delay of lipid metabolism lacks a comparison with key studies on lipid synthesis.
Response 4: We are not entirely clear as to what the reviewer is asking here as it may relate to either fatty acid biosynthesis, or suberin specific modifications (e.g., elongation, oxidation, etc.), or indeed both. If we interpret the comment in relation to wound-induced suberization, based on our previous work measuring soluble fatty acids (Yang & Bernards Plant Signaling Behav. 2006, 1 (2), 59–66. https://doi.org/10.4161/psb.1.2.2433), and transcript abundance of genes associated with fatty acid biosynthesis (Woolfson et al. Phytochemistry 2023, 206, 113529. https://doi.org/10.1016/j.phytochem.2022.113529.) there is little or no lipid biosynthesis during the early stages of wound healing. The main explanation may lie in the differential regulation of genes by ABA, but this is speculative. Nevertheless, we have added new text to the discussion to address this comment (lines 577-584 in the revised manuscript).
Comment 5: In the Materials and Methods section, the key instrument settings for GC - MS are missing, such as the inlet temperature, ion source temperature, and scanning mode.
Response 5: These details have been added.
Comment 6: The manufacturers of some reagents are not marked, such as "triacontane internal standard".
Response 6: We have added a new section 4.2 “Chemicals and Reagents” to specify the source and quality of reagents used (lines 637-652 in the revised manuscript). This replaces explicit catalogue numbers within the text.